# Complexities of Type I Interferon Biology: Lessons from LCMV

**DOI:** 10.3390/v11020172

**Published:** 2019-02-20

**Authors:** Tamara Suprunenko, Markus J. Hofer

**Affiliations:** School of Life and Environmental Sciences, the Marie Bashir Institute for Infectious Diseases and Biosecurity, Charles Perkins Centre, and the Bosch Institute, The University of Sydney, Sydney, NSW 2006, Australia; tsup5163@uni.sydney.edu.au

**Keywords:** lymphocytic choriomeningitis virus, type I interferon, innate immune response, adaptive immune response, CD8^+^ T cells, pathogen recognition, chronic infection, acute infection

## Abstract

Over the past decades, infection of mice with lymphocytic choriomeningitis virus (LCMV) has provided an invaluable insight into our understanding of immune responses to viruses. In particular, this model has clarified the central roles that type I interferons play in initiating and regulating host responses. The use of different strains of LCMV and routes of infection has allowed us to understand how type I interferons are critical in controlling virus replication and fostering effective antiviral immunity, but also how they promote virus persistence and functional exhaustion of the immune response. Accordingly, these discoveries have formed the foundation for the development of novel treatments for acute and chronic viral infections and even extend into the management of malignant tumors. Here we review the fundamental insights into type I interferon biology gained using LCMV as a model and how the diversity of LCMV strains, dose, and route of administration have been used to dissect the molecular mechanisms underpinning acute versus persistent infection. We also identify gaps in the knowledge regarding LCMV regulation of antiviral immunity. Due to its unique properties, LCMV will continue to remain a vital part of the immunologists’ toolbox.

## 1. Introduction

From MHC recognition to functional T cell exhaustion, lymphocytic choriomeningitis virus (LCMV) has been central to the discovery of a wide range of ground-breaking immunological concepts. Its non-cytopathic nature makes it an ideal model as disease is not caused by the virus itself, but the subsequent antiviral immune response [1]. This feature has made LCMV the virus of choice for many immunologists that study the immune response to viruses, allowing researchers to tease apart the contribution of virus versus host response in immune pathology. A key component of this antiviral host response are the type I interferons (IFN-Is) and much of what is known about their biology comes from studies using LCMV in mouse models. The IFN-I family, which includes several IFN-αs and a single IFN-β amongst other subtypes, are rapidly produced following infection with viruses including LCMV, and play essential roles in activating and modulating innate and adaptive immunity. In this review we will outline the fundamental immunological insights into IFN-I biology gained using LCMV as a model.

## 2. Lymphocytic Choriomeningitis Virus (LCMV): One Virus with Many Outcomes

Over 30 strains of LCMV have been described since the original isolation of LCMV-Armstrong in 1933 [2]. However only six of these are routinely used in research; Armstrong (Arm), Clone-13 (Cl13), Traub, WE, Aggressive and Docile (summarized in Table 1). These laboratory strains originate from three parental strains. The isolates LCMV-Arm 53b (often referred to as LCMV-Arm) and LCMV-Cl13 both originate from a common ancestor LCMV-Arm strain [3]. The second strain, LCMV-Traub, was isolated by Erich Traub from a persistently infected mouse in 1935 [4]. The third parental strain is the WE strain from which a number of sub-strains have been derived [5]. The WE strain has a somewhat unclear history. It was originally isolated from a patient in 1935 and, following the transfer of virus stocks in the 1940s, became known as LCMV-UBC [6]. However, coding changes between LCMV-WE and LCMV-UBC suggest that both strains are independent sub-strains (Figure 1). Two additional sub-strains were isolated from an LCMV-UBC carrier mouse, LCMV-Docile and LCMV-Aggressive [7]. Furthermore, multiple isolates of LCMV-WE have been characterized, including the clones WE c54 and WE c2.2 [8].

Importantly these strains differ significantly in their biology and pathogenicity in mice and, together with dosage and route of administration, the virus strain has a significant impact on host response and the outcome of infection (Table 1). For example, acute infection is achieved using LCMV-Arm and injecting 1 × 10^3^ to 3 × 10^5^ pfu (plaque forming units) intraperitoneally (IP) or intracranially (IC). By contrast, chronic infection can be induced following intravenous (IV) injection of 2 × 10^6^ pfu LCMV-Cl13. The differences in strain, dose and route of infection are also reflected in the IFN-I response, with IFN-α and IFN-β responses following LCMV-Cl13 infection being several fold higher compared with LCMV-Arm infection when using above regimens [9]. However, when infected with the same dose IV, LCMV-Arm elicits a stronger and more prolonged IFN-α response than LCMV-Cl13 [10]. Waggoner et al. further demonstrated that different doses of LCMV-Cl13 (5 × 10^4^ pfu, 2 × 10^5^ pfu and 2 × 10^6^ pfu) lead to distinct outcomes, with the middle dose resulting in >20% mortality rate, whereas all mice survive when infected with the low or high dose [11]. Even the genetic background of mice influences IFN-I responses and viral persistence [12]. For example, C57BL/6 mice produce much more IFN-I compared with Sv129 mice in response to LCMV [13]. The distinct array of pathogenic outcomes that arise from the impact of virus strain, dosage and route of administration is a double-edged sword. While they broaden our perception of the complexity of host–virus interactions, they also hinder our ability to construct a larger picture of the antiviral response, as results must be interpreted in a context-dependent manner.

Infection of C57BL/6 mice with LCMV-Arm via the IP route results in an acute and innocuous infection that is cleared within 10–12 days. Clearance of LCMV is mediated by virus specific CD8^+^ T cells [27,28] and is dependent on IFN-I signaling [16,19,29] but largely independent of CD4^+^ T cell help [30,31]. By contrast, IC infection with LCMV-Arm causes a lethal neurological disease, mediated by a cytotoxic CD8^+^ T cell response [18]. This disease is characterized by sudden onset of seizures and T cell infiltrates in the meninges and choroid plexus. Accordingly, this disease has been termed lymphocytic choriomeningitis (LCM).

By contrast, LCMV-Cl13 is frequently used to establish a chronic infection in mice through high doses administered via an IV route. Importantly, chronic LCMV-Cl13 infection results in an incapacitated CD8^+^ T cell response, termed functional or T cell exhaustion. CD8^+^ T cell exhaustion was first described 20 years ago by Zajac et al. in LCMV-Cl13 infected mice [32]. In this setting, virus specific CD8^+^ T cells become functionally incapacitated and are unable to produce key antiviral cytokines or lyse virus-infected cells (see [33,34] for a review of functional exhaustion). There is also a distinct molecular signature present in exhausted CD8^+^ T cells, which includes an increased expression of co-inhibitory receptors, such as programmed death-1 (PD-1), lymphocyte activation gene-3 (LAG-3), T cell immunoglobulin and mucin-domain containing 3 (TIM-3) and cytotoxic T-lymphocyte–associated antigen 4 (CLTA-4), amongst others. Induction of exhaustion is not limited to high-dose IV injection of LCMV-Cl13 as it can also be induced following infection with other LCMV strains using specific regimens [19]. For example, while low-dose LCMV-Docile is cleared from wild-type (WT) mice, a high dose results in viral persistence and CD8^+^ T cell exhaustion [19]. Since the original study by Zajac et al. using LCMV, T cell exhaustion has been extensively studied in numerous mouse models for a wide range of infectious and non-infectious diseases. These studies have subsequently revealed the importance of T cell exhaustion in a number of human diseases including chronic viral infections, such as human immunodeficiency virus (HIV), hepatitis B virus (HBV) and hepatitis C virus (HCV), as well as a broad range of malignant tumors including melanoma and ovarian cancer. Moreover, our understanding of the molecular mechanisms driving T cell exhaustion, which were in large parts established in the LCMV mouse model, has led to the development of novel therapies targeting co-inhibitory receptors, in particular the PD-1 pathway (reviewed in [33,35,36]).

In addition to dose and route of administration, genetic differences between virus strains have a significant impact on clinical manifestation following infection. The six commonly used strains of LCMV show as much as 21% difference on the protein level (Figure 1). There are two coding changes in LCMV-Cl13 compared with LCMV-Arm that contribute to its persistence. The first is a lysine to glutamine (K1079Q) substitution in the viral RNA polymerase (L protein) resulting in enhanced intracellular replication [37]. The second is a phenylalanine to leucine (F260L) mutation located in the glycoprotein (GP) that results in a high affinity for the surface receptor, α-dystroglycan. This mutation allows LCMV-Cl13 to preferentially infect dendritic cells (DCs) [38,39,40]. Infection of DCs with LCMV reduces their expression of antigen-presenting and costimulatory molecules and thus impairs their ability to prime T cells [40]. As a consequence, these mutations allow LCMV-Cl13 to overwhelm the host and create an environment that promotes viral persistence [41]. A third change (N176D in GP) was not found to influence viral persistence of LCMV-Cl13 [37].

Similar to LCMV-Arm, LCMV-WE and LCMV-Aggressive are slowly replicating virus strains and cause an acute infection in mice that is cleared within two weeks. In contrast, the more aggressively replicating strains LCMV-Traub and LCMV-Docile result in a persistent infection [16,19] (Table 1), suggesting that virus load is one of the key factors in establishing virus persistence. Despite the prevalence of these strains in research, there is little published on how genetic differences between strains affect the outcome of infection. Similar to LCMV-Cl13, in addition to faster replication, increased affinity for the entry receptor α-dystroglycan also contributes to virus persistence. For example, leucine or isoleucine at position 260 in the virus GP, which is present in LCMV-Cl13, Docile and Aggressive, increase affinity to α-dystroglycan [16,39,42]. Yet, additional mutations in the GP appear to aid LCMV in establishing chronic infection. The two sub-strains of LCMV-WE, c54 and c2.2, cause different outcomes of infection despite both isolates possessing a leucine at GP position 260. Both sub-strains have different affinities for α-dystroglycan, which is probably due to a single amino acid substitution at position 153 in the GP [39,42,43]. Similarly, the K1079Q mutation associated with enhanced replication in LCMV-Cl13 is conserved between LCMV-Arm and all other virus strains examined, even those that cause persistent infection [37], highlighting that clearance or persistence of LCMV infection cannot be attributed to one or two amino acid substitutions alone.

## 3. Detection of LCMV and Induction of the Antiviral Response

Host recognition of a pathogen is dependent on pathogen recognition receptors (PPRs) recognizing pathogen-associated molecular patterns (PAMPS). In the context of LCMV, there are two pathways that come into play; the Toll-like receptors (TLRs) and the retinoic acid inducible gene 1 (RIG-I)-like receptors (RLRs). The expression of these two classes of PPRs differs with the TLRs being found primarily on macrophages and DCs, while RLRs are widely expressed in most cell types [44]. Unlike most other enveloped viruses, LCMV enters the cell via the endosome [45], where it is detected by the endosomal TLRs, TLR-7 and -8 that respond to viral single-stranded (ss)RNA. Of note TLR3, which also contributes to virus detection, recognizes double-stranded (ds)RNA. Yet, it does not appear to be involved in the response to LCMV [46,47]. Conversely, the membrane-bound TLR2 has been implicated in the response to LCMV. Typically, TLR2 triggers proinflammatory cytokine production via the NF-κB in response to viral proteins. While TLR2 is not required for eliminating LCMV-Arm or LCMV-WE, it is needed for a normal production of interleukin (IL)-6 and the chemokine CCL2 [48]. Furthermore, Zhou et al. show that TLR2 plays an important role in glial cells in the central nervous system (CNS), where it is required for chemokine production [47]. In addition to the proinflammatory response, TLR2 has been shown to contribute to IFN-I production, with mice deficient for TLR2 having significantly reduced bioactivity of IFN-I in the serum following LCMV-WE infection [48].

All TLRs with the exception of TLR3 signal through the adaptor protein MyD88. Ligation of the receptor and activation of MyD88 results in a complex signaling cascade that culminates in the activation and translocation to the nucleus of the transcription factors AP-1 and NF-κB inducing gene transcription of proinflammatory cytokines such as tumor necrosis factor (TNF) and IL-12 (reviewed in [49]). This signaling pathway is utilized by macrophages, conventional DCs (cDCs), and plasmacytoid DCs (pDCs) (Figure 2). However, TLRs expressed on the endosome of pDCs are also able to signal through an additional unique pathway allowing for the production of large amounts of IFN-I, a hallmark of pDCs [50]. In pDCs, signaling via MyD88 and the kinases IRAK-1 and IKKα results in the phosphorylation of the constitutively expressed transcription factor interferon regulatory factor (IRF) 7, which stimulates expression of IFN-Is (reviewed in [49,51]).

In contrast to the endosomal PRRs TLR7 and TLR8, the RLRs melanoma differentiation-associated protein (MDA)-5 and RIG-I detect nucleic acids in the cytosol. Specifically, they recognize dsRNA and 5’-triphosphate RNA, respectively, that are formed as part of the LCMV replication cycle [52]. Binding of PAMPs to MDA-5 and RIG-I stimulates their ATPase/helicase activity resulting in the exposure of the caspase recruitment domain (CARD), which binds to the adaptor protein mitochondrial antiviral signaling protein (MAVS (aka IPS-1, VISA, CARDIF)). A signaling complex then forms containing TRAF3, TBK1 and IKKε, which activates the transcription factor IRF3 (Figure 3). IRF3 acts together with AP-1 and NF-κB to induce the expression of the early IFN-I species IFN-β [53] and IFN-α_4_ [54]. In parallel, NF-κB stimulates the expression of proinflammatory cytokines (reviewed in [55]).

Once produced, the early IFN-I species signal in an autocrine and paracrine manner to induce the expression of interferon-stimulated genes (ISGs), including that of *Irf7* [54,56] (Figure 3). Unlike IRF3 which is constitutively and ubiquitously expressed, it is generally accepted that IRF7 is either not expressed or expressed at very low levels in resting cells [50,57]. Subsequently, IRF7 is activated in a similar fashion to the MAVS-IRF3 pathway resulting in its phosphorylation at the C-terminus and homo- or heterodimerization with IRF3 [57]. The IRF7:IRF7 and IRF7:IRF3 complexes then translocate to the nucleus and induce the expression of the other IFN-α subtypes [54]. This two-step induction of the early and secondary IFN-I production is responsible for amplification and the tight regulation of the IFN-I response (Figure 3).

Like many other viruses, LCMV has evolved strategies to counteract its recognition by PRRs and the induction of IFN-I expression. The viral nucleoprotein (NP) directly associates with RIG-I and MDA-5 to inhibit the induction of IFN-Is [52]. In addition, the NP protein can also block the translocation of IRF3 to the nucleus, thereby significantly reducing the production of IFN-β [58]. Yet, this inhibition is incomplete as evident from the robust production of IFN-Is in mice infected with LCMV and the initiation of a potent antiviral host response [9].

In mice, the IFN-I family consists of 14 subtypes of IFN-α [59], IFN-β as well as the lesser well characterized subtypes IFN-κ, -ω, -ε and -ζ (reviewed in [60,61]). This review will focus on the IFN-αs and IFN-β. All IFN-I subtypes signal through the same heterodimeric transmembrane receptor, comprised of IFNAR1 and IFNAR2 which is present on most nucleated cells (Figure 3). Importantly, each of the IFN-I subtypes has different affinities for the receptor resulting in differences in downstream signaling and cellular effects (reviewed in [62]). Also, the coordinated expression of multiple subtypes is required for a normal and effective response. For example, blockade of IFN-β alone does not alter virus dissemination during the first 24 h following LCMV-Cl13 infection, whereas complete IFN-I signaling deficiency increases virus replication [63], indicating that despite IFN-β being produced first, IFN-α is also critical for mediating the very early antiviral effects against LCMV. Following the binding of IFN-Is, the receptor chains dimerize, activating the receptor-associated tyrosine kinases, Janus kinase 1 (JAK1) and tyrosine kinase 2 (TYK2) (Figure 3). Once activated, the tyrosine kinases trans-phosphorylate each other as well as tyrosine residues in the cytoplasmic tails of the receptor chains. This allows docking of the transcription factors signal transducer and activator of transcription (STAT) 1 and STAT2 at the receptor and their subsequent tyrosine phosphorylation by JAK1 and TYK2. Once phosphorylated, STAT1 and STAT2 dissociate from the receptor and together with IRF9 form the trimolecular interferon-stimulated gene factor 3 (ISGF3) complex, which then translocates to the nucleus [64,65] in order to stimulate expression of several hundred ISGs.

In addition to the ISGF3 complex, which is an integral part of IFN-I signaling, non-canonical signaling via the IFNAR receptor contributes to the overall cellular response to IFN-Is (reviewed in [66,67,68]). Although much progress has been made towards identifying the nature of these non-canonical signaling pathways, their exact role in the IFN-I response remains unclear. For example, studies using LCMV infection of mice deficient of either STAT1, STAT2 or IRF9 revealed distinct outcomes of infection suggesting that a coordinated activation of different pathways is critical for a normal host immune response (see below for more detail and [69]).

## 4. Type I Interferons (IFN-Is) are More than a Mediator of the Innate Response

Since their discovery more than 60 years ago, our understanding of IFN-Is has expanded tremendously and numerous studies have shown that IFN-Is are required for the innate as well as the adaptive immune response against viral infection. In many aspects, the LCMV model has been instrumental in these discoveries. For example, the seminal study by Müller and colleagues describing the generation of IFN-I receptor-deficient mice utilized LCMV amongst other viruses to demonstrate the central role that IFN-Is play in antiviral host responses *in vivo* [29]. The production of IFN-Is in response to LCMV-Arm or -Cl13 infection is detectable as early as 6 h post-infection, peaking within the first 24 h [9,10,13]. This rapid increase in IFN-I levels results in the production of several hundred ISGs that mediate and regulate the antiviral host response and whose expression parallels that of the LCMV titre [70].

### 4.1. The Role of IFN-Is in the Outcome of LCMV Infection Varies Between Virus Strains

The importance of IFN-Is in controlling the LCMV has been clearly demonstrated. Loss of IFN-I signaling either by blockade or deletion of the receptor IFNAR results in chronic infection and increased sensitivity to slowly replicating strains of LCMV including Arm, WE, Aggressive and a low dose of LCMV-Docile that would be otherwise cleared by WT mice [16,19,29,71]. In IFNAR-deficient mice, which are deficient in all responses to IFN-Is, LCMV-Arm persistence is accompanied by increased replication and the virus spreads to all organs including the CNS [9,72,73]. This is not limited to the LCMV model but a comparable need for IFN-Is in controlling virus replication is seen for many other viruses (for examples refer to [29,74,75]). Similarly, a recently reported patient with a mutation in the *IFNAR2* gene, rendering them unresponsive to IFN-Is, was highly susceptible to otherwise innocuous viruses and succumbed to lethal encephalitis following vaccination with a live attenuated measles-mumps-rubella vaccine [76]. Comparable to peripheral infection, IC infection of IFNAR-deficient mice with LCMV-Arm results in generalized virus spread, whereas LCMV is largely contained in the CNS of similarly infected WT mice (unpublished observations Suprunenko and Hofer). However, unlike WT mice, which develop classical LCM and die around 7–8 days post infection, IFNAR-deficiency is protective in the IC infection model and IFNAR-deficient mice or mice treated with neutralizing antibodies to IFN-I are mostly free from signs of disease [29,73,77]. Yet this comes at the cost of virus persistence and chronic infection ([73] and unpublished observations Suprunenko and Hofer). Although mice treated with anti-IFN-I serum are protected from overt pathological changes in the CNS following IC infection with LCMV-Arm, there are distinct pathological changes in the lymph nodes and spleen [73], indicating that inflammation and tissue pathology in LCMV-infected animals is not a result of antigen levels but of the host immune response. In contrast to LCMV-Arm and other slowly replicating LCMV strains, virus persistence is not affected by IFN-I signaling deficiency following infection with more aggressively replicating strains of LCMV, such as Traub, Docile and Cl13. Yet, while persistence is seen in both WT mice and mice that lack IFN-I signaling, viral titres are significantly increased in mice that lack IFN-I signaling early during infection [9,70].

Importantly, in contrast to LCMV-Arm where IFN-Is are critical in preventing persistent infection, during LCMV-Cl13 infection signaling via IFNAR has been implicated in facilitating viral persistence and promoting exhaustion of the CD8^+^ T cell response. Whilst loss of IFNAR1 or IFNAR1 blockade initially increases LCMV-Cl13 titres [9,70], suggesting that IFNAR blockade cancels the antiviral effects of IFN-I effectively, by day 30 post infection virus levels are reduced compared with mice that did not receive blockade [9,70]. Moreover, blockade of IFNAR counteracts the immunosuppressive environment induced by LCMV-Cl13, with reduced expression of the co-inhibitory receptor PD-1 and the anti-inflammatory cytokine IL-10 following blockade of IFNAR1 [9,70]. Yet, reduced virus load and enhanced viral clearance in these mice is not due to increased CD8^+^ cytotoxic T cell activity but is mediated by an enhanced CD4^+^ T cell response and IFN-γ-dependent mechanisms [9,70].

While a lot has been learnt from dampening the IFN-I response such as by the works of Wilson et al. and Teijaro et al. and many others, studies have also been undertaken in mice with augmented IFN-I responses. Administration of recombinant IFN-I during the first week post infection can prevent exhaustion of the CD8^+^ T cell response and establishment of LCMV-Cl13 chronic infection [10]. A similar effect has been observed in mice with IFN-I blockade prior to infection [9,70]. However, IFN-I administered after the first week of LCMV-Cl13 infection [10,78] has no impact on the outcome of infection. Together these studies show that the actions of IFN-Is can dramatically differ depending on the timing. Thus, IFN-I-mediated clearance of LCMV-Cl13 coincides with the priming of the CD8^+^ T cells [10], suggesting that IFN-Is play a critical role in initiating an effective adaptive immune response. Similarly, the ability to clear LCMV-Cl13 infection can also be achieved by “priming” mice with LCMV-Arm prior to LCMV-Cl13 infection. Mice that are primed between 4 and 8 h prior to infection with LCMV-Cl13 are able to clear LCMV-Cl13 infection within two weeks [79]. This is due to priming-triggered IFN-α production downstream of RIG-I- and MDA-5-dependent viral RNA sensing. Consequently, a robust CD8^+^ T cell response is established that results in the mice clearing LCMV-Cl13 infection [79].

The groundbreaking work on the importance of timing of the IFN-I response and its impact on the outcome of infection has assisted in the development of IFN-I therapies to treat chronic viral infections and some cancers in humans. This includes the use of pegylated interferon in HBV and HCV infection, where it is often used in combination with other drugs, such as IL-10 or antiviral agents (reviewed in [80]). Blockade of IFN-I as a therapeutic approach is still being investigated in animal models, where it has shown mixed outcomes in terms of therapeutic benefit. The blockade of IFN-Is in simian immunodeficiency virus infection leads to enhanced viral replication, whereas anti-IFNAR2 treatment of HIV infection in humanized mice increases anti-HIV CD8^+^ T cell responses (reviewed in [81]). Similarly, clinical trials of IFN-I therapy in cancer have produced mixed results. The use of IFN-α was standard therapy to treat chronic myelogenous leukemia up until 2001 when tyrosine kinase inhibitors became the primary treatment [82]. However, clinical trials have been looking at combination therapy with IFN-α to improve responses to tyrosine kinase inhibitors; however, dosages need to be optimized to avoid toxicity [83]. In contrast, studies in breast and ovarian cancer have reported limited improvement and IFN-Is are associated with toxicity (reviewed in [81,84]). These mixed outcomes of IFN-Is in therapy mirror that of what we have learnt from the LCMV model; that whether IFN-Is are beneficial or detrimental to the host is highly dependent on the specific context. While the LCMV model may not have been the sole basis for the development of these therapeutic approaches, it has provided invaluable insights and understanding necessary for the subsequent development of IFN-Is as drugs.

### 4.2. IFN-Is Modulate the Innate Immune Response by Regulating Interferon-Stimulated Gene (ISG) Expression and Innate Immune Cells

The two main components of the innate immune system that mediate antiviral effects are a vast array of soluble mediators and the activation of innate immune cells such as natural killer (NK) cells and monocytes. Surprisingly little detail is known regarding the mechanisms of how IFN-Is affect the innate immune response against LCMV. Despite the rapid induction of ISGs in response to IFN-I signaling, their contribution to controlling LCMV infection is not well understood [85]. For example, Lee et al. found that mice that lack the ISG 2’-5’ oligoadenylate synthetase-like 1 (OASL1) have an enhanced CD8^+^ T cell response that is capable of overcoming LCMV-Cl13 infection [86]. The enhanced survival of the CD8^+^ T cells in OASL1-deficient mice is dependent on higher IFN-I production early following infection (days 2–3 post-infection), which is the consequence of loss of negative regulation of IRF7 by OASL1 [86,87]. This is in line with the findings of Wang et al., who reported that administering IFN-Is during the first week of infection improved CD8^+^ T cell responses against LCMV-Cl13 [10].

In addition to stimulating the expression of a large number of ISGs, IFN-I play a role in activating cells of the innate immune response such as NK cells. Although, the NK cell response to LCMV is not well characterized, they appear to have no significant role in the antiviral response against LCMV-Arm, with NK cell depletion having minimal impact on viral burden [88]. However, in the context of chronic LCMV infection, NK cells have an inhibitory effect on T cell responses, whereby the NK cells eliminate activated CD4^+^ T cells in a perforin-dependent manner, and consequently affect CD8^+^ T cell function through the loss of CD4^+^ T cell help [11]. Thus, depletion of NK cells prior to LMV-Cl13 infection results in enhanced effector function of CD4^+^ and CD8^+^ T cells [11,88], leading to a significant reduction of viral titres during the later stages of infection [88]. This effect is dependent on IFN-Is. Xu et al. found that in the absence of IFN-Is, NK cells target antiviral T cells through perforin-mediated toxicity [89], demonstrating that IFN-Is protect T cells from NK cell-mediated toxicity, at least in the context of LCMV infection. Notably, this was done in the context of LCMV-WE infection, which can be resolved in WT mice but persists in IFNAR-deficient mice [19]. These findings highlight not only how the role of IFN-Is varies during the innate immune depending on the type of infection, but also emphasizes the need to consider virus strain-specific differences when interpolating findings.

### 4.3. IFN-Is are Central Regulators of T and B Cell Activity

Adaptive immunity is mediated primarily by antigen-specific T and B cells, which rely on IFN-Is for their activation and differentiation into effector cells. In the context of LCMV infection, IFN-Is provide an important signal to support CD8^+^ T cell expansion and differentiation [90,91]. The absence of IFNAR signaling results in CD8^+^ T cells from LCMV-Arm-infected mice displaying signs of classic exhaustion coinciding with an inability to clear the virus [29,72]. Importantly, intrinsic loss of IFNAR expression on CD8^+^ T cells does not render the cells defective. These cells are still able to proliferate following antigen stimulation, express activation markers, and produce cytokines in response to LCMV-Arm and LCMV-WE [90,91]. It is the T cell extrinsic loss of IFNAR, that renders LCMV-specific CD8^+^ T cells exhausted [72]. While these cells are able to proliferate in response to antigens, the majority of their progeny die as they are unable to respond to survival signals provided by direct IFN-I signaling [72,90,91]. In addition to CD8^+^ effector T cells, long-lasting memory T cells are also produced during antigen-driven CD8^+^ T cell expansion. Similar to the effector response, survival of memory CD8^+^ T cells is dependent on IFN-I signaling [91] and a lack thereof may impact on the host’s ability to clear a secondary infection.

The importance of IFN-Is in modulating the CD8^+^ T cell response is highlighted in the interplay between IFN-Is and IL-12. The cytokine IL-12 plays an important role in promoting T cell function [92]. In the context of infection with other pathogens, IL-12 has a role in the CD8^+^ T cell response. However, during LCMV infection IL-12 does not contribute CD8^+^ T cell expansion or function under normal circumstances [92,93]. This is likely the consequence of the high levels of IFN-Is that are produced following infection of immunocompetent mice with LCMV and which suppress IL-12 production. Consequently, in the absence of IFNAR, IL-12 levels are increased [94]. While the increase of endogenous IL-12 in IFN-I signaling-deficient LCMV-infected mice can promote the antiviral state, it cannot completely compensate for the loss of endogenous IFN-Is [93]. Yet, when IL-12 is administered as an adjuvant, CD8^+^ T cells deficient for IFNAR can overcome the reduced expansion caused by intrinsic loss of IFNAR [91]. These studies demonstrate the extensive role that IFN-Is play in immunomodulation, a capacity that is unable to be compensated for in the absence of IFN-I signaling.

Helper CD4^+^ T cells play important roles in stimulating and regulating the activity of immune responses. They do this mainly by producing various cytokines, which modulate the activity of immune cells. Although, the CD4^+^ T helper cell response is not known to contribute significantly in the resolution of acute LCMV-Arm infection in WT mice [9,95,96], IFN-Is are involved in their activation and differentiation. Havenar-Daughten et al. have shown that in the absence of IFNAR on CD4^+^ T cells, the CD4^+^ T cells are still able to become activated and proliferate following infection with LCMV-Arm; however, similar to CD8^+^ T cells, IFN-Is are required for their survival during the antigen-driven proliferation stage [97]. While CD4^+^ T cell help is not required to activate CD8^+^ T cell responses, it is required to maintain CD8^+^ T cell responses during persistent infection such as with LCMV-Cl13 [32,96,98]. In line with this, enhanced control of LCMV-Cl13 in mice that lack IFN-I signaling is dependent on CD4^+^ T cells [9,70]. Further, in the absence of IFN-I signaling, primary CD4^+^ T cells show improved expansion in response to infection with LCMV-Cl13, however it is not currently known whether the immunoregulatory properties of IFN-Is on CD4^+^ T helper cells is in a cell intrinsic or extrinsic fashion [99]. Notably, the absence of IFN-Is alters the differentiation of CD4^+^ T cells, with bias towards a follicular B helper (Tfh) phenotype over a cell-mediated immune response promoting Th1 phenotype, resulting in improved antibody responses [99]. IFN-Is also have an immunoregulatory effect on regulatory T cells (Tregs) in a cell intrinsic manner during LCMV-Arm infection [100]. Tregs play an important role in modulating T cell responses during infection and prevent excessive expansion of effector T cells. However, they can also regulate effector function of T cells and consequently, loss of INFAR in Tregs results in impairment of CD4^+^ and CD8^+^ T cell responses and compromised viral clearance following LCMV-Arm infection [100]. In contrast to the defective CD4^+^ and CD8^+^ T cells responses in LCMV-infected IFNAR-deficient mice, IRF7-deficient mice exhibit normal T cell responses [101,102]. This discrepancy may be due to the complete absence of all IFN-I signaling in the first model, but residual signaling and responses in the latter. While IRF7-deficient mice have undetectable bioactivity of IFN-I during early infection [102], they are still capable of producing normal levels of IFN-β [101].

The third cell type of the adaptive immune system are B cells. They are important antigen presenting cells and form the second arm of the adaptive immune response - humoral immunity. Compared with T cell responses much less is known about the contribution of B cells to anti-LCMV responses (reviewed in [81]). Likely this is an intrinsic consequence of the non-cytopathic nature of LCMV with the antiviral response dominated by T cells versus that of cytopathic viruses such as vesicular stomatitis virus (VSV) which elicits a strong neutralizing antibody response during the early stages of infection (reviewed in [103]). Thus, B cells are not required for the generation of the T cell response or early control of acute LCMV-Arm infection [104]. This differs in the context of a persistent infection, where the immune response is more complex, requiring both T and B cell mechanisms to control viremia [105,106]. B cells also play a role in promoting optimal T cell responses, as in the absence of B cells the CD8^+^ T cell response is impaired following infection with the fast replicating LCMV-Traub [107]. Interestingly, this defect is more pronounced in mice with a BALB/C background than those on a C57BL/6 background [107], highlighting the impact of host genetic background on disease outcomes.

In accordance with their central role in regulating immunity, the effects of IFN-Is extend to B cell responses. Moseman et al. have demonstrated that IFN-Is can facilitate CD8^+^ T cell killing of B cells. During IFNAR blockade, cytotoxic CD8^+^ T cells are exhausted and deletion of B cells is prevented resulting in an accelerated development of neutralizing antibodies, which in turn enhances the antiviral response against LCMV-Cl13 [108]. However, this is not a direct impact of IFN-Is on the B cells but rather a consequence of the immunoregulatory functions of IFN-Is on T cells. Moreover, Daugan et al. showed that chimeric mice with disrupted IFN-I signaling in B cells had an enhanced antibody response to LCMV-Cl13 compared with WT mice [109]. Interestingly, the effect was less pronounced in LCMV compared with VSV, despite LCMV eliciting stronger IFN-I responses, suggesting LCMV-specific factors contribute to this effect [109]. Together, these studies demonstrate that IFN-Is play direct and indirect roles in the humoral response to LCMV infection.

In summary, from these studies, it is evident that IFN-Is can either be beneficial or harmful to the host depending on a variety of factors, including the timing of the IFN-I response, virus strain and route of infection. Studies using LCMV as a model have clearly shown that in addition to having a direct role in activating the innate immune response, IFN-Is are also critical for inducing an effective CD8^+^ T cell response and modulating CD4^+^ T helper responses and antibody production.

## 5. A Coordinated Induction of IFN-Is is Critical for Early Antiviral Responses Against LCMV

While a number of studies have focused directly on IFN-I signaling by targeting the receptor, or blocking or adding specific IFN-I species, work has also been done on the pathways that lead to the production of IFN-Is following infection with LCMV. This includes the recognition of LCMV by the PPRs, as well as the induction of IFN-I production through IRF3 and IRF7 signaling.

The contribution of the RLR and TLR pathways in LCMV has been controversial and differs in the context of acute versus chronic infection. Of note, the relative contribution of each pathway has been studied mostly by focusing on the respective adaptor proteins, MAVS and MyD88, and consequently the cell types that express them. At the center of these studies are the DCs. DCs possess a central role in sensing viral pathogens and initiating T cell responses thereby linking innate and adaptive immunity. DCs can be divided up into subsets, cDCs and pDCs. The pDCs are able to rapidly produce large amounts of IFN-Is during the first 24 h of LCMV infection [110] via IRF7, in an MyD88 signaling-dependent manner [110]. Although pDCs are critical for the immediate early IFN-I response to LCMV, Cervantes-Barragan et al. found that mice lacking pDCs are able to mount a CD8^+^ T cell response and clear LCMV-Arm infection [111]. While the RLR pathway is not involved in the immediate early production of IFN-I by pDCs, IFN-Is produced by cDCs in response to LCMV-Arm is dependent on MAVS signaling and peaks at day 2 post-infection [52,110]. Reduced IFN-I levels in mice that are deficient for MAVS coincide with increased viral titres and defective T cell responses to both LCMV-Arm and LCMV-Cl13 [44]. This suggests that both pDCs and cDCs are important for controlling virus replication and elimination. Similarly, mice deficient in MyD88 fail to control acute infection and are unable to mount an effective CD8^+^ T cell response following infection with LCMV-Arm or LCMV-WE [44,48,52,110,112]. Importantly, this is independent of signaling through viral-PAMP recognizing TLRs, TLR-3, -7, -9, since IFN-I production, early virus control and T cell responses to acute infection are largely unaffected in Unc3b1 mice (3d mice) that lack all three TLRs [44]. However, in response to chronic infection, 3d mice have a defective antibody response and T cell responses diminish over time [44], indicating that during infection TLR-dependent virus recognition is important for a sustained immune response. Rather than being dependent on TLRs and an impaired pDC response, the inability of MyD88 deficient mice to mount an effective CD8^+^ T cell response is due to direct effects of MyD88 on T cells [48,112]. This MyD88 T cell intrinsic role is required for the survival of CD8^+^ T cells [112].

In contrast, pDCs are indispensable for the response to persistent infection. During LCMV-Docile infection, the absence of pDCs in mice results in an a defective CD4^+^ T cells response and consequently a defective CD8^+^ T cell reaction [111]. This is likely due to the absence of IFN-Is, required for proper activation of the T cell response, rather than the antigen presentation capacity of the pDCs, as mice that lack the antigen presentation molecule MHCII on pDCs have normal CD8^+^ T cells [111]. However, to elucidate specifically the contribution of IFN-Is produced by pDCs, an experimental model in which the secretion of IFN-Is from pDCs alone is abolished, is needed [111].

The increased expression of IFN-Is in response to recognition of LCMV by PRRs is mediated by signaling pathways that activate the transcription factors IRF3 and IRF7. As discussed above, IRF7 is an essential factor in the amplification of IFN-I production, and IRF7 but not IRF3 is required for the induction of IFN-Is following IP infection with LCMV-Arm [52]. Consequently, mice deficient for IRF7 are more susceptible to viral infection than their WT counterparts. While early viral replication is unable to be controlled, IRF7-deficient mice eventually mount an effective CD8^+^ T cell response and are capable of clearing LCMV-Arm following IP infection [101,102]. This is likely due to an impaired antiviral state, as the production of IFN-α, but not IFN-β, is dependent on IRF7 [101,113,114]. Likewise, absence of IRF7 slightly delays the IFN-I response in the CNS [113], thereby the onset of LCM following IC infection with LCMV-Arm [101]. Together these studies demonstrate that while IRF7 is not essential for the IFN-I response to LCMV, it is required for the optimal early antiviral response to infection.

## 6. Efficient Antiviral Responses to LCMV Require Canonical IFN-I Signaling Through the ISGF3 Complex

The ISFG3 complex, which comprises STAT1, STAT2 and IRF9 and is activated in response to IFN-I signaling, is responsible for mediating the majority of the cellular effects of IFN-Is. Accordingly, infection of mice lacking components of the ISGF3 complex show abnormal immune responses to LCMV. However, deficiency of each component of the ISGF3 complex has distinct consequences for the outcome of infection with LCMV-Arm [69,72]. Following IP infection with LCMV-Arm, STAT1-deficient mice develop a lethal wasting disease. This disease is accompanied by increased virus replication and spread, and multiorgan tissue destruction [69,115] and increased IRF7-dependent IFN-I levels [115,116]. Interestingly, IC infection of STAT1-deficient mice results in the same wasting disease rather than classical LCM [77], suggesting that development of classical LCM relies on multiple factors including controlled virus replication and spread, and a precisely tuned antiviral host response. By contrast, the wasting disease depends on non-canonical STAT1-independent IFN-I signaling through STAT2 and IRF9 [116], highlighting the dual role for IFN-Is in being beneficial or harmful to the host, depending on the context. In contrast to WT mice, where the CD8^+^ T cell response mediates clearance of LCMV following IP infection or causes LCM following IC infection, these cells are not critically involved in the immunopathology and weight loss observed in LCMV-Arm-infected STAT1-deficient mice [69]. By contrast, CD4^+^ T cells are essential for the wasting disease, yet the exact nature of this pathogenic CD4^+^ T cell response remains unclear.

In contrast to STAT1-deficient mice [117], mice that lack STAT2 do not develop lethal disease after IP or IC infection with LCMV-Arm, despite both mice showing uncontrolled viral replication and spread [69,77,115]. Instead, STAT2-deficient mice develop only moderate clinical signs accompanied by inflammatory foci in peripheral organs. In addition to affecting the outcome of infection, STAT2 also regulates DC expansion. Following infection with LCMV-Cl13, DCs produce large amounts of IFN-Is, which in an autocrine manner activate STAT2 resulting in the suppression of DC expansion. This is independent of STAT1 [118], providing further evidence for STAT1-independent, STAT2-dependent IFN-I signaling.

The third component of the ISGF3, IRF9, is the main DNA binding component of the transcription factor complex, providing the specificity for binding to gene promoter regions containing the interferon-stimulated response element (ISRE) [119,120,121]. Thus, it is not surprising that IRF9-deficiency renders cells incapacitated to respond to IFN-Is [122,123,124]. Accordingly, IRF9-deficient mice behave similar to IFNAR-deficient mice and survive both IP and IC infection with LCMV-Arm [69,72]. Likewise, IRF9-deficient mice also show increased susceptibility to a range of other viruses, including VSV and herpes simplex virus (HSV) [124] that is also seen in IFNAR-deficient mice [29,117]. While infection of IRF9-deficient mice with LCMV comes at the cost of persistent viral infection and inflammatory foci in kidney, liver and lungs, there is only minor tissue damage and mice recover from signs of disease by day 14 post-infection [69,72]. In contrast to the STAT1-deficient mice, which show exaggerated IFN-I levels [116], IRF9 is required for optimal IFN-I production. Reduced IFN-I levels in the absence of IRF9 are likely due to reduced *Irf7* gene expression and consequently disruption of the IFN-I amplification loop [72]. Similar to mice lacking IFNAR, IRF9-deficient mice also develop functional exhaustion of LCMV-specific CD8^+^ T cells [72]. Transfer of IRF9-sufficient and -deficient virus specific CD8^+^ T cells into WT or IRF9-deficient mice has demonstrated that the incapacitated T cell response is mediated by CD8^+^ T cell extrinsic mechanisms. Together these studies show that ISGF3-independent signaling not only exists but plays important roles not just in regulating gene expression but also in modulating antiviral immune responses.

## 7. Conclusions

The non-cytopathic nature of LCMV combined with the ability to manipulate disease outcomes through strain diversity, dose and route of administration have made it an ideal model to dissect the molecular mechanisms underpinning acute versus persistent infection. Here, we have highlighted the distinct roles of IFN-Is, from induction of the antiviral response, to their immunomodulatory properties on the adaptive immune response. Nevertheless, much remains to be discovered regarding LCMV regulation of antiviral immunity. As a result, LCMV will continue to be instrumental in illuminating the diverse nature of anti-viral responses and will provide an important basis for the development of novel immunotherapy approaches.

## Figures and Tables

**Figure 1 viruses-11-00172-f001:**
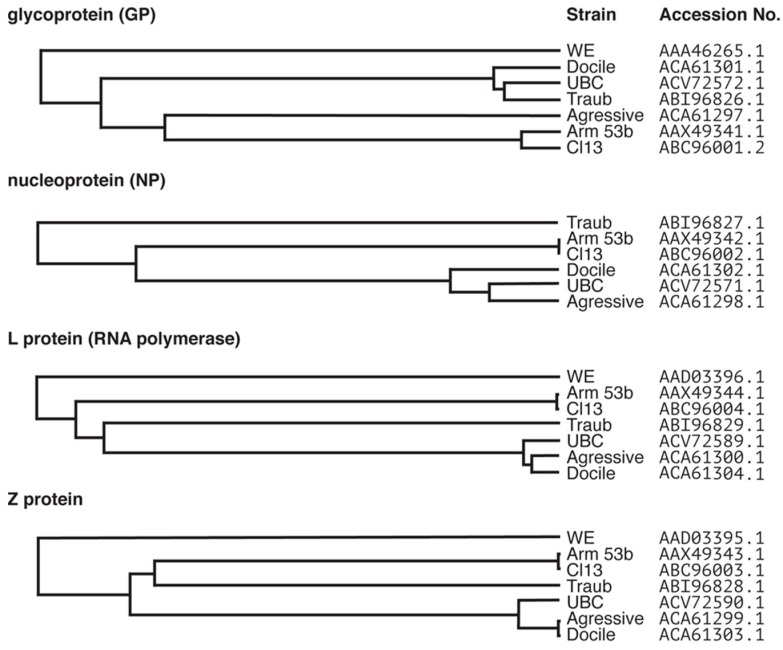
Clustal Omega dendrogram predications generated from the alignments of protein sequences of common laboratory strains of LCMV. The LCMV genome consists of the S and L segments. The S segment encodes the glycoprotein (GP) and the nucleoprotein (NP) and the L segment encodes the L protein (RNA polymerase) and the Z protein. Protein sequences for commonly used laboratory strains LCMV-Armstrong 53b, Clone13, Traub, UBC, WE, Aggressive and Docile were aligned using Clustal Omega and dendrograms generated from the alignments. Sequences were obtained from Genbank^®^ and accession numbers used for alignment are indicated. Note, there was no sequence available for the NP of LCMV-WE. The identity between the alignments is as follows; GP: 90.361%, NP: 94.086%, L protein: 81.315%, Z protein: 78.889%.

**Figure 2 viruses-11-00172-f002:**
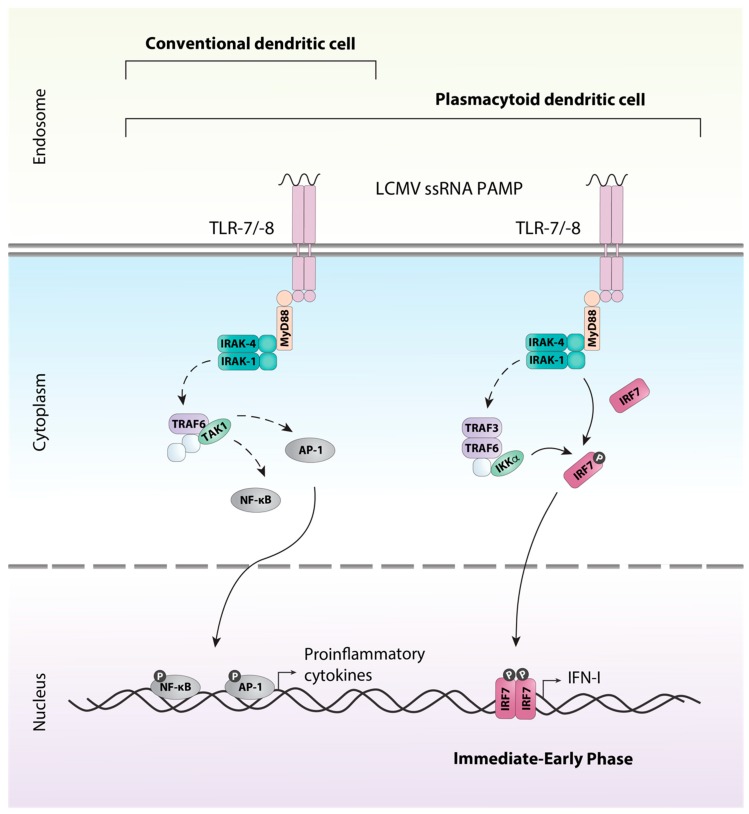
TLR-7/-8 signaling in dendritic cells. LCMV ssRNA is detected by the endosomal TLRs; TLR7 and TLR8, expressed by conventional dendritic cells (cDCs) and plasmacytoid dendritic cells (pDCs). The association of MyD88 with the receptor results in the association of IRAK1 and IRAK4 through the death domains. Once phosphorylated the IRAK proteins dissociate from the receptor to associate with TRAF6. TRAF6 interacts directly and indirectly with a number of other proteins, including protein kinases such as TAK1. TAK1 can then triggers the activation of transcription factors such as AP-1 via the MAPK pathway or NF-κB through the canonical IKK complex. Upon their activation, the transcription factors translocate to the nucleus and induce production of inflammatory cytokines. In addition, pDCs utilize a unique pathway that allows for the rapid immediate-early production of type I interferons. Similar to above, MyD88 associates with the receptor and a complex including TRAF3, TRAF6, IRAK1, IRAK4 and IKKα is formed. IRAK1 and IKKα phosphorylate the transcription factor IRF7, resulting in its activation and subsequent production of type I interferon (IFN-I). Solid arrows indicate direct steps, dashed arrows indicate multiple steps/additional proteins involved.

**Figure 3 viruses-11-00172-f003:**
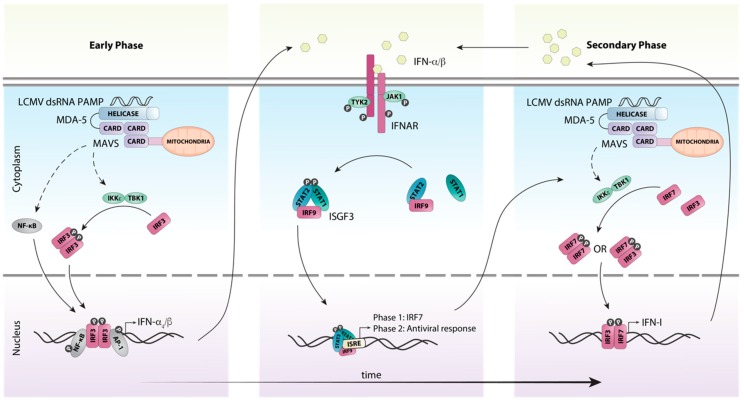
Schematic representation of host detection of LCMV and subsequent type I interferon induction. The production of IFN-Is in response to LCMV infection can be categorized into early and secondary phases. Upon recognition of viral dsRNA by the pattern recognition receptor MDA-5, the adaptor protein MAVS is recruited, initiating a signaling cascade involving multiple factors. This cascade results in the phosphorylation and subsequent activation of the transcription factor IRF3 by the kinases IKKε and TBK1. Simultaneously, the transcription factors NF-κB and AP1 are also activated and translocated to the nucleus. Together these transcription factors initiate the transcription of the initial IFN-I species IFN-β and IFN-α_4_. These early IFN-Is subsequently bind to the IFN-I receptor IFNAR in an autocrine or paracrine fashion and activate canonical IFN-I signaling through the ISGF3 complex. This results in the transcription of interferon-stimulated genes, including the transcription factor IRF7. IRF7 is then activated in a similar fashion to IRF3, resulting in the amplification of the secondary IFN-I response. Solid arrows indicate direct steps, dashed arrows indicate multiple steps/additional proteins involved.

**Table 1 viruses-11-00172-t001:** Origin and outcome of infection of the six commonly used lymphocytic choriomeningitis virus (LCMV) strains.

Strain	Origin	Replication and Pathogenicity ^1^
Armstrong 53b	Originally isolated from an infected patient in 1993 [14] and obtained from a triple plaque purified clone and two passages in BHK-21 (baby hamster kidney) cells [15].	Slowly replicating. Peripheral infection causes acute infection; virus clearance within 2 weeks from wild-type (WT) mice [3,16]. Intracranial infection results in lethal lymphocytic choriomeningitis (LCM) [17,18].
Clone-13	13th isolate of Armstrong that differed from the parental strain in that it persisted in mice [3]	Faster replicating than LCMV-Arm 53b. Peripheral infection with high dose virus via intravenous route causes persistence, whereas low doses are cleared [3,16].
Traub	Isolated from persistently infected mouse [4]	Faster replicating. Peripheral infection results in chronic infection that is cleared within 2–4 months [16,19]. Intracranial infection causes LCM [20].
WE ^2^	Isolated from an infected patient (WE) with meningo-encephalitis in in 1935 [5]	Slowly replicating. Peripheral infection is cleared within 2 weeks [16,19,21,22]. Mice survive intracranial infection ([23] as reviewed in [22])
Aggressive	Isolated from an LCMV-WE [UBC] carrier mouse [7] Note, UBC is a derivative of WE [24]	Slowly replicating. Peripheral infection is cleared within 2 weeks [16]. Intracranial infection with low doses causes LCM, however high dose infected mice survive [25]
Docile	Isolated from an LCMV-WE [UBC] carrier mouse [7] Note, UBC is a derivative of WE [24]	More quickly replicating than parent strain. Peripheral infection results in persistent infection [16,19,22]. Intracranial infection with high dose results in lifelong persistence, but low dose infected mice survive [25]

^1^ Outcome is dependent on route and dose of infection and genetic background of mouse strain. For details refer to original publications; ^2^ The history of LCMV-WE is somewhat unclear; however, all publications mentioned in this review refer to the original isolate (unless specified otherwise) and can be traced to Lehmann-Grube, Hamburg, Germany [26].

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
