# Peer review of "Complexities of Type I Interferon Biology: Lessons from LCMV"

_viruses, 2019, doi:10.3390/v11020172_

Reviewer 1 Report

This focused and authoritative review details how LCMV biology has shaped our understanding of interferon biology (and vice versa).  The review is generally well written, timely, and comprehensive.  Some suggestions that could improve the value of this review follow, including some important comments about accessibility for a less well-versed reader:

Major

1.  Line 42:  the authors correctly note that multiple strains of LCMV are used experimentally, and describe some aspects of each in the text that follows.  It might be worth summarizing in a table how the six used in research differ in terms of sequence and overall pathogenic outcomes following infection in mice.  While some of this is covered in the text, to my knowledge, no resource that directly contrasts these distinct strains exists, and this could help to address some confusion in the literature.

2.  The review faithfully documents IFN biology following LCMV infection; one question that sometimes arose as I read the review was whether LCMV was the first virus used to identify this principle, and/or whether other viruses behave in a similar way, or that LCMV is unique in its response.  For a reader who may be seeking to learn more about IFN biology following viral infections in general, how much of a prototype LCMV has been would be informative.

3.  The sections are quite long (eg, section 4) and address multiple and varied topics.  Perhaps including sub-sections (or more sections) would provide clear indication when the topic will shift.

4.  There are occasions (eg, para beginning on 294) where the content is more a listing of facts than a statement of a over-arching concept with supporting data.  This strategy may make it harder for the novice to LCMV to see the forest from the trees.  More decisive intro sentences (as well as more clarity about sections; #7 above) might help provide a helpful skeleton of the review for such readers.

Minor

5. Line 99:  “…prime T cells through”—not sure what the original intent was; this needs to be edited.

6.  Line 205:  change to “IFN-a is also critical for mediating…”

7.  Line 272:  the phrase “8 to 4 hours…” seemed odd.  Please check.

8.  Paragraph beginning on line 277:  This is interesting, certainly, but was LCMV critical for the development of hypotheses related to anti-cancer approaches (other than the proof-of-principle provided in line 291)?

Author Response

We thank the reviewer for their positive comments and helpful critique. We have incorporated their comments and suggestions in the revised manuscript. Please find below a point-by-point response.

1.  …It might be worth summarizing in a table how the six [strains] used in research differ in terms of sequence and overall pathogenic outcomes following infection in mice.

We agree with the reviewer that such a summary and comparison would increase the value of the manuscript. Thus, we have included a table (table 1) as well as a figure (new Figure) in the revised manuscript outlining the differences between LCMV strains and the outcome in mice following infection.

2.  … one question that sometimes arose as I read the review was whether LCMV was the first virus used to identify this principle, and/or whether other viruses behave in a similar way … . For a reader … how much of a prototype LCMV has been would be informative.

We have included statements where appropriate to clarify this. E.g. lines 258-261, 335-337 and 531-533.

3.  The sections are quite long (eg, section 4) and address multiple and varied topics.  Perhaps including sub-sections (or more sections) would provide clear indication when the topic will shift.

We agree with the reviewer and have included subheadings in section 4. See lines 266,338 and 366.

4.  There are occasions (eg, para beginning on 294) where the content is more a listing of facts than a statement of a over-arching concept with supporting data.  …  More decisive intro sentences …might help provide a helpful skeleton of the review for such readers.

We thank the reviewer for this suggestion and have included subheadings (see also point 3 above), and summary and introductory statements where relevant, e.g. 147-148, 339-341, 394-396, 420-421, and 453 -454.  

5. Line 99:  “…prime T cells through”—not sure what the original intent was; this needs to be edited.

 This has been corrected (line 119 of revised manuscript).

6.  Line 205:  change to “IFN-a is also critical for mediating…”

 This has been corrected (line 237 of revised manuscript).

7.  Line 272:  the phrase “8 to 4 hours…” seemed odd.  Please check.

 This has been corrected (line 313 of revised manuscript).

8.  Paragraph beginning on line 277:  This is interesting, certainly, but was LCMV critical for the development of hypotheses related to anti-cancer approaches (other than the proof-of-principle provided in line 291)?

This has been clarified in the revised manuscript (lines 334-336 of the revised manuscript).

Reviewer 2 Report

this is an excellent review focused on the roles of Type I interferons in antiviral responses. The manuscript provides extensive overview on the complex interaction of type I IFNs in relation to LCMV viral strains, dose and route of infection. Exemples are adequatly chosen to perfectly illustrate the authors's points of view.

Author Response

We thank the reviewer for their very positive comments and their time to read our manuscript.